# ABOUT THE ATTRACTOR PHENOMENON IN DECOMPOSED REINFORCEMENT LEARNING

**Romain Laroche, Mehdi Fatemi, Joshua Romoff & Harm van Seijen**
MSR Montréal, Canada
romain.laroche@microsoft.com

## ABSTRACT

We consider tackling a single-agent RL problem by decomposing it to $n$ learners. These learners are generally trained *egocentrically*: they are greedy with respect to their own local focus. In this extended abstract, we show theoretically and empirically that this leads to the presence of attractors: states attracting and detaining the agent, against what the global objective function would advise.

## 1   INTRODUCTION

When a person faces a complex and important problem, his individual problem solving abilities might not suffice. He has to actively seek for advice around him: he might consult his relatives, browse different sources on the internet, and/or hire one or several people that are specialised in some aspects of the problem. He then aggregates the advice in order to hopefully make the best possible decision. A large number of papers tackle the decomposition of a single Reinforcement Learning task (RL, Sutton & Barto, 1998) into several simpler ones. They generally follow a method where agents are trained independently and greedily with respect to their local optimality, which we call *egocentric*, and where their recommendations are aggregated into a global policy by voting or averaging.

Unlike Hierarchical RL (Dayan & Hinton, 1993; Parr & Russell, 1998; Dietterich, 2000), this approach gives the learners the role of advisors. The learners are said to have a *focus*: reward function, state space, learning technique, etc. This approach endeavours therefore to first, independently tackle these different focuses and afterwards, merge their advice. Section 2 shows that the *egocentric* planning presents the severe theoretical shortcoming of inverting a $\max \sum$ into a $\sum \max$ in the global Bellman equation. It leads to an overestimation of the values of states where the learners disagree, and creates an *attractor* phenomenon, causing the system to remain static without any tie-breaking possibilities. We show that attractors can be avoided by lowering the discount factor $\gamma$. Section 3 illustrates and empirically validates the theoretical results.

## 2   THEORETICAL ASPECTS

**Markov Decision Process** – The Reinforcement Learning (RL) framework is formalised as a Markov Decision Process (MDP). An MDP is a tuple $\langle \mathcal{X}, \mathcal{A}, P, R, \gamma \rangle$ where $\mathcal{X}$ is the state space, $\mathcal{A}$ is the action space, $P : \mathcal{X} \times \mathcal{A} \to \mathcal{X}$ is the Markovian transition function, $R : \mathcal{X} \times \mathcal{A} \to \mathbb{R}$ is the reward function, and $\gamma$ is the discount factor. The goal is to generate trajectories with high discounted cumulative reward, also called more succinctly *return*: $\sum_{t=0}^{T-1} \gamma^t r(t)$. To do so, one needs to find a policy $\pi : \mathcal{X} \times \mathcal{A} \to [0, 1]$ maximising the $Q$-function:
$$Q_\pi(x, a) = \mathbb{E}_\pi \left[ \sum_{t' \geq t} \gamma^{t'-t} R(X_{t'}, A_{t'}) | X_t = x, A_t = a \right].$$

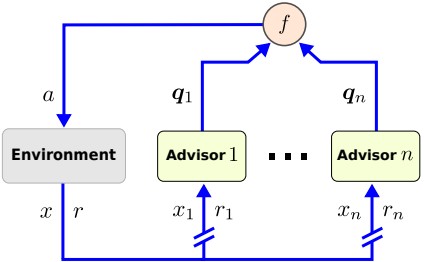

Figure 1: The architecture

In our setting, we assume that the task has been broken down into $n$ learners, which are regarded as specialised, possibly weak, learners that are concerned with a sub part of the problem. The overall architecture is illustrated in Figure 1. At each time step, learner $j$ sends to the aggregator

its local $Q$-values for all actions in the current state. The $f$ function's role is to aggregate the learners' recommendations into a policy. For the further analysis, we restrict ourselves to the linear decomposition of the rewards: $R(x, a) = \sum_j w_j R_j(x_j, a)$. We define the aggregator function $f_\Sigma(x)$ as being greedy over the $Q_j$-functions aggregation $Q_\Sigma(x, a)$.

We recall the theoretical result of van Seijen et al. (2017b): a theorem ensuring, under the local Markov condition, that the learners' training eventually converges. Although it guarantees convergence, it does not guarantee the optimality of the converged solution.

***Egocentric* planning** – It is the most common approach in the literature Singh & Cohn (1998); Russell & Zimdars (2003); Harutyunyan et al. (2015). Theorem of van Seijen et al. (2017b) guarantees for each learner $j$ the convergence to the local optimal value function, denoted by $Q_j^{ego}$, which satisfies the Bellman optimality equation:

$$Q_j^{ego}(x_j, a) = \mathbb{E}\left[r_j + \gamma \max_{a' \in \mathcal{A}} Q_j^{ego}(x_j', a')\right],$$

where the local immediate reward $r_j$ is sampled according to $R_j(x_j, a)$, and the next local state $x_j'$ is sampled according to $P_j(x_j, a)$. In the aggregator global view, we get:

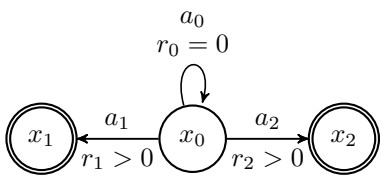

$$Q_\Sigma^{ego}(x, a) = \mathbb{E}\left[\sum_j w_j r_j + \gamma \sum_j w_j \max_{a' \in \mathcal{A}} Q_j^{ego}(x_j', a')\right]$$

$$\geq \mathbb{E}\left[r + \gamma \max_{a' \in \mathcal{A}} Q_\Sigma^{ego}(x', a')\right].$$

Figure 2: Attractor example.

*Egocentric* planning suffers from an inversion between the $\max$ and $\sum$ operators and, as a consequence, it overestimate the state-action values when the learners disagree on the optimal action. This flaw has critical consequences in practice: it creates *attractor* situations. Before we define and study them formally, let us explain attractors with an illustrative example based on the simple MDP depicted in Figure 2. In initial state $x_0$, the system has three possible actions: stay put (action $a_0$), perform learner 1's goal (action $a_1$), or perform learner 2's goal (action $a_2$). Once achieving a goal, the trajectory ends. The $Q$-values for each action are easy to compute: $Q_\Sigma^{ego}(s, a_0) = \gamma r_1 + \gamma r_2$, $Q_\Sigma^{ego}(s, a_1) = r_1$, and $Q_\Sigma^{ego}(s, a_2) = r_2$. As a consequence, if $\gamma > r_1/(r_1 + r_2)$ and $\gamma > r_2/(r_1 + r_2)$, the local *egocentric* planning commands to execute action $a_0$ endlessly.

**Definition 1.** *An attractor $x$ is a state where the following strict inequality holds:*

$$\max_{a \in \mathcal{A}} \sum_j w_j Q_j^{ego}(x_j, a) < \gamma \sum_j w_j \max_{a \in \mathcal{A}} Q_j^{ego}(x_j, a).$$

**Theorem 1.** *State $x$ is attractor, if and only if the optimal egocentric policy is to stay in $x$ if possible.*

Note that there is no condition in Theorem 1 (proofs are omitted because of space constraint) on the existence of actions allowing the system to be actually static. Indeed, the system might be stuck in an attractor set, keep moving, but opt to never achieve its goals. To understand how this may happen, just replace state $x_0$ in Figure 2 with an attractor set of similar states: where action $a_0$ performs a random transition in the attractor set, and actions $a_1$ and $a_2$ respectively achieve tasks of learners 1 and 2. Also, it may happen that an attractor set is escapable by the lack of actions keeping the system in an attractor set. For instance, in Figure 2, if action $a_0$ is not available, $x_0$ remains an attractor, but an unstable one.

**Definition 2.** *A learner $j$ is said to be progressive if the following condition is satisfied:*

$$\forall x_j \in \mathcal{X}_j, \forall a \in \mathcal{A}, \quad Q_j^{ego}(x_j, a) \geq \gamma \max_{a' \in \mathcal{A}} Q_j^{ego}(x_j, a').$$

The intuition behind the progressive property is that no action is worse than losing one turn to do nothing. In other words, only progress can be made towards this task, and therefore non-progressing actions are regarded by this learner as the worst possible ones.

**Theorem 2.** *If all the learners are progressive, there cannot be any attractor.*

The condition stated in Theorem 2 is very restrictive. Still, there exist some RL problems where Theorem 2 can be applied, such as resource scheduling where each learner is responsible for the progression of a given task. Note that a setting without any attractors does not guarantee optimality for the *egocentric* planning. Most of RL problems do not fall into this category. Theorem 2 neither applies to RL problems with states that terminate the trajectory while some goals are still incomplete, nor to navigation tasks: when the system goes into a direction that is opposite to some goal, it gets into a state that is worse than staying in the same position.

**Navigation problem attractors** – We consider the three-fruit attractor illustrated in Figure 3: moving towards a fruit, makes it closer to one of the fruits, but further from the two other fruits (diagonal moves are not allowed). The expression each action $Q$-value is as follows: $Q_{\Sigma}^{ego}(x,S) = \gamma \sum_j \max_{a \in \mathcal{A}} Q_j^{ego}(x_j, a) = 3\gamma^2$, and $Q_{\Sigma}^{ego}(x, N) = Q_{\Sigma}^{ego}(x, E) = Q_{\Sigma}^{ego}(x, W) = \gamma + 2\gamma^3$. That means that, if $\gamma > 0.5$, $Q_{\Sigma}^{ego}(x,S) > Q_{\Sigma}^{ego}(x, N) = Q_{\Sigma}^{ego}(x, E) = Q_{\Sigma}^{ego}(x, W)$. As a result, the aggregator would opt to go South and hit the wall indefinitely.

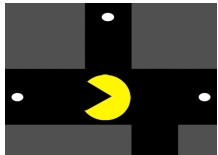

Figure 3: Attractor.

More generally in a deterministic task where each action $a$ in a state $x$ can be cancelled by a new action $a_x^{-1}$, it can be shown that the condition on $\gamma$ is a function of the size of the action set $\mathcal{A}$.

**Theorem 3.** *State $x \in \mathcal{X}$ is guaranteed not to be an attractor if:*

$$\begin{cases} \forall a \in \mathcal{A}, \exists a_x^{-1} \in \mathcal{A}, \text{ such that } P(P(x,a), a_x^{-1}) = x \,, \\ \forall a \in \mathcal{A}, R(x,a) \geq 0 \,, \\ \gamma \leq \frac{1}{|\mathcal{A}|-1} \,. \end{cases}$$

## 3 PAC-BOY EXPERIMENT

Figure 4: The Pac-Boy game.

In this section, we empirically validate the findings of Section 2 in the Pac-Boy domain van Seijen et al. (2017a): a fruit collection task (see Figure 4). The decomposed setting is associating one learner for each of the 75 potential fruit locations. The local state space consists in the agent position and the existence –or not– of the fruit. Four different settings are compared: the two baselines *linear Q-learning* and *DQN-clipped*, and *egocentric* with $\gamma = 0.4$, *egocentric* with $\gamma = 0.9$. The *linear Q-learning* and *egocentric* with $\gamma = 0.9$ do not succeed at getting positive rewards. *DQN-clipped* reaches a 25-reward average, while *egocentric* with $\gamma = 0.4$ approaches the 37.5 optimal reward, with an average of 36.

One can notice that the Markov assumption holds in this setting and that, as a consequence, the theorem from van Seijen et al. (2017b) applies. Theorem 3 determines sufficient conditions for not having any attractor in the MDP. In the Pac-Boy domain, the cancelling action condition is satisfied for every $x \in \mathcal{X}$. As for the $\gamma$ condition, it is not only sufficient but also necessary, since being surrounded by goals of equal value is an attractor if $\gamma > 1/3$. In practice, an attractor becomes stable only when there is an action enabling it to remain in the attraction set. Thus, the condition for not being stuck in an attractor set can be relaxed to $\gamma \leq 1/(|\mathcal{A}|-2)$. Hence, the result of $\gamma > 1/2$ in the example illustrated by Figure 3.

We provide links to 3 video files (click on the blue links) representing a trajectory generated at the 50[th] epoch for various settings. *egocentric-$\gamma = 0.4$* adopts a near optimal policy coming close to the ghosts without taking any risk. The fruit collection problem is similar to the travelling salesman problem, which is known to be NP-complete (Papadimitriou, 1977). However, the suboptimal small-$\gamma$ policy consisting of moving towards the closest fruits is in fact a near optimal one. Regarding the ghost avoidance, *egocentric* with small $\gamma$ gets an advantage over other settings: the local optimisation guarantees a perfect control of the system near the ghosts. The most interesting outcome is the presence of the attractor phenomenon in *egocentric-$\gamma = 0.9$*: Pac-Boy goes straight to the centre area of the grid and does not move until a ghost comes too close, which it still knows to avoid perfectly. This is the empirical confirmation that the attractors present a real practical issue. Finally, we observe that *DQN-clipped* struggles to eat the last fruits.

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
