# OpenReview forum: "About the attractor phenomenon in decomposed reinforcement learning"
_ICLR.cc/2018/Workshop — Reject_

### Official Review · AnonReviewer3 · 2018-03-10
**slightly sloppy take on a problem of growing interest**

**Rating:** 4
**Confidence:** 3

**Review:**

I've been noticing more and more people talking about these sorts of problems---merging the advice of multiple policies. There was a lot of work along these lines back in the olden days and, I believe, people abandoned these approaches because there wasn't a well reasoned way to combine results and know whether the combination would perform well.

It's worth revisiting these issues, but I'd really like to see some sort of discussion about what might be different this time around. This paper (probably in part due to its brevity!) doesn't really provide sufficient context to convey why it's worth considering such approaches again. It also gets kind of sloppy at the end, with incomplete sentences and missing words here and there. I don't think the paper is ready yet.

---

### Official Review · AnonReviewer4 · 2018-03-18
**Unsurprising paper with no take home message**

**Rating:** 5
**Confidence:** 3

**Review:**

This paper shows that for a high enough discount factor, policies that follow a linear combination of egocentric policies have an attractor which could be quite suboptimal. The examples used in the paper are simple and the result is not surprising. The authors also lack motivation of using such a simple linear combination policy, making the paper borderline towards rejection.
In summary, this paper states a problem which is obvious (greedy/sub-optimal policies can pull you towards opposite directions), without a clear path forward.

---

### Decision · Program_Chairs · 2018-03-20
**ICLR 2018 Workshop Acceptance Decision**

**Decision:**

Reject

**Comment:**

Based on the reviews, this paper has not been accepted for presentation at the ICLR workshop. However, the conversation and updates can continue to appear here on OpenReview.